# Prevalence of atrial fibrillation in Northern Sri Lanka: a study protocol for a cross-sectional household survey

Shribavan Kanesamoorthy [1], Vethanayagam Antony Sheron,[1]
Powsiga Uruthirakumar,[1] Chamira Kodippily,[2] Balachandran Kumarendran,[1]
Tiffany E Gooden,[3] Graham Neil Thomas [3], Krishnarajah Nirantharakumar [3],
Gregory Y H Lip [3,4] Mahesan Guruparan,[5] Rashan Haniffa,[6]
Rajendra Surenthirakumaran,[1] Abi Beane,[6] Kumaran Subaschandran,[1] on behalf of
the NIHR Global Health Research Group on Atrial Fibrillation Management

For numbered affiliations see end of article.

**Correspondence to**
Professor Graham Neil Thomas; gneilthomas@gmail.com

## ABSTRACT

**Introduction** Atrial fibrillation (AF) is the most common arrhythmia globally. It is associated with a fivefold risk in stroke, but early diagnosis and effective treatment can reduce this risk. AF is often underdiagnosed, particularly in low-income and middle-income countries (LMICs) where screening for AF is not always feasible or considered common practice in primary care settings. Epidemiological data on AF in LMICs is often incomplete particularly in vulnerable populations. This LMIC research collaborative aims to identify the prevalence of AF in the Northern Sri Lankan community.

**Methods and analysis** A cross-sectional household survey piloted and codesigned through a series of community engagement events will be administered in all five districts in Northern Province, Sri Lanka. A multistage cluster sampling approach will be used starting at district level, then the Divisional Secretariats followed by Grama Niladhari divisions. Twenty households will be selected from each cluster. The study aims to recruit 10 000 participants aged 50 years or older, 1 participant per household. Demographic and socioeconomic characteristics, well-being and lifestyle and anthropometric measurements will be collected using a digital data platform (REDCap, Research Electronic Data Capture) by trained data collectors. Participants will be screened for AF using a fingertip single-lead ECG via a smartphone application (AliveCor) with rhythm strips reviewed by a consultant cardiologist. Prevalence of AF and risk factors will be established at province and district-levels. Adjusted ORs and population attributable fractions for AF risk factors will be determined.

**Ethics and dissemination** This study was approved by the Ethics Review Committee of Faculty of Medicine at University of Jaffna. Written informed consent will be obtained from all participants. Findings will be disseminated through publication in a peer-reviewed journal and presentations at conferences. The findings will enable early treatment for new AF diagnoses and inform interventions to improve community-based management of AF in LMICs.

### STRENGTHS AND LIMITATIONS OF THIS STUDY

⇒ This study uses mHealth (mobile health) technology that has the ability to quickly and efficiently identify atrial fibrillation (AF) within resource-limited settings.
⇒ The use of a near-real time data collection tool and analytical dashboard will enable the ability to identify and rectify any data collection issues as the study progresses.
⇒ Conducting the study in this low-income and middle-income countries setting will inform future non-communicable screening policies to extend the evidence base beyond high-income settings.
⇒ While data pertaining to a variety of risk factors will be collected, the temporal relationship between risk factors and AF will not be attained.
⇒ Given the cross-sectional study design, incidence of AF and other non-communicable diseases will not be feasible.

## INTRODUCTION

Atrial fibrillation (AF) is the most common cardiac rhythm disorder globally.[1] AF prevalence is increasing worldwide with age, male sex, hypertension, diabetes and heart disease as major risk factors.[2] AF is associated with a fivefold increase in stroke and over 25% of the strokes are due to AF.[3] Early identification and effective management of AF can prevent strokes and contribute to improved management of cardiovascular health.[3–5] Stroke caused by AF is largely preventable with appropriate use of oral anticoagulant therapy, a therapy which is readily available in health systems internationally.[5] However, an estimated 33% of the people with AF are asymptomatic, meaning many people with AF remain undiagnosed until they present with life threatening complications.[6]

Screening for AF in people aged 65 years or older is now recommended by some clinical practice guidelines and is common place in general practice in upper-middle and high-income countries.[7 8] Such population screening has in part been made possible by the successful development and scale up of smartphone technologies, which enable rapid point of care diagnosis through opportunistic pulse palpation or ECG rhythm strip.[9 10]

Such practice remains limited in low-income and middle-income countries (LMICs), where cardiovascular diseases are being increasingly reported in younger populations (aged 50 years and above), and which are rapidly attributing to causes of mortality.[11–14] Chronic disease prevalence in this relatively young population appears as a result of both childhood conditions such as rheumatic fever and valvular heart disease, and early onset of diabetes and hypertension as a result of age and lifestyle-related factors.[11 15] Additionally, complications associated with non-communicable diseases are attributed with worse outcomes and greater morbidity for people living in LMICs when compared with high-income settings.[16] Despite this disparity in diagnosis, management and outcomes, there is limited high quality epidemiological data on AF in LMICs, specifically in South Asia.[2 15]

Screening programmes in the regions are hindered by poor recruitment, often as a result of poor community engagement and low levels of non-communicable disease-related health literacy. Where programmes do exist, vulnerable groups are often under-represented and missing or incomplete data inhibits meaningful interpretation of data.[17] Epidemiology researchers are increasingly looking to social science and implementation strategies to engage communities in research design to ensure they are included in such programmes. Particularly in LMICs, it is imperative to overcome these known barriers to study implementation and ensure that study findings are not only disseminated back to communities, but are both acceptable to and used by healthcare providers.

This study aims to implement an AF screening programme using smartphone technology, feasible for delivery at scale in resource constrained LMIC settings. Specifically, we aim to determine the prevalence of AF and associated risk factors in Northern Province, Sri Lanka, and to evaluate the feasibility and acceptability of smartphone-enabled single-lead ECG for the purpose of diagnosing AF.

## METHODS AND ANALYSIS
### Research design
This cross-sectional study combines a household questionnaire and an onsite assessment of the presence of AF using a single-lead ECG reader on a digital mobile health (mHealth) platform. The study commenced in June 2020 and is expected to be complete by March 2022.

### Study setting
This study is being conducted across all five districts in the Northern Province of Sri Lanka: Jaffna, Kilinochchi, Mannar, Mullaitivu and Vavuniya. Each district is divided into administrative units called Divisional Secretariats (DS). Each DS is further divided into many Grama Niladhari (GN) divisions. Each GN division is identified by a unique number. Northern Province is located just 22 miles (35 km) southeast of India and has approximately 1.3 million permanent residents.[18] Its climate is typically tropical, but with a hot dry season from February to September. While the Northern Province was heavily impacted by 30 years of conflict, the province has recently opened to the global market and its share of provincial gross domestic product is increasing.[19] Publicly available healthcare including medical appointments, medications and medical procedures are free to all Sri Lankan citizens.

### Sampling and sample size
Consistent with other LMIC studies and in response to the increasing literature on morbidity and chronic disease prevalence in South Asia, we selected a threshold of 50 years and above for inclusion in screening, a lower than current recommended practice of 65 years.[12] Therefore, individuals aged 50 years or above will be considered eligible; additionally, they must be fluent in the Tamil language. Based on already published methods,[20] anyone suffering from terminal illnesses, in need of immediate hospital admission or currently an inpatient in a hospital setting will be excluded. All data collectors—which includes a medically trained doctor and two nursing graduates—will be trained to assess individuals for eligibility.

This study will use a multistage cluster sampling approach. The multistage process will start at the district level, then to the DS level followed by the GN level. GN divisions will then be divided into clusters based on population size and from these clusters, a single cluster will be selected at random. Each cluster will consist of 20 households, one participant per household. We will first randomly select an index house within each cluster and from the index house, 20 households on the right side will be selected. In households where more than one eligible person lives, the person with a birthday closest to the date of visit will be selected. We used 2012 population census data to select the clusters and index house within each cluster.[19]

For the sample size calculation, the prevalence of AF in Sri Lanka was estimated to be 1% based on previous evidence.[2 21 22] We used a design effect of 2 to adjust for cluster sampling, and we used an alpha level of 5% and beta level of 20%. We increased our sample size by 10% to account for non-participation,[23] leading to a minimum sample size of 10 000 participants required.

### Pilot study
The data collection tool was piloted in 100 households located in a GN division that will not be included within the main study. The pilot study comprised participants

from varying socioeconomic backgrounds and education levels. Participants were asked to give their perspectives on acceptability of participating in the study including the duration of the interview, ease of understanding the interviewer administered questionnaire and their experience of using the fingertip rhythm assessment ECG application. Additionally, data collectors fed back on the challenges they experienced in collecting the data and reported on their perception of participants' willingness to participate and recall, internal consistency reliability and inter-rater reliability of the questionnaire. During the piloting of the questionnaire, participants' questions and concerns were noted and reported through the electronic data collection tool used; these were reviewed daily by the principal and co-investigators.

Feedback following the pilot revealed (1) data collector concerns regarding availability of and access to clinical books for relevant medical history, (2) participant concerns regarding future use of fingerprints, reasons for the survey and being disadvantaged if they were found to have AF and (3) participants understanding of medical prescriptions. In response, we made the following adaptations to the study conduct protocol prior to commencing data collection: reporting of medical history will be reported as either being directly observed from clinical books or via participant recall; medications will be reported by categories for simplicity and; an update of participant information leaflets to ensure the aim of the study, data security and what happens after the study is clear. Additionally, further training for the data collection team was provided on the management of personal information, the AliveCor device and how to respond to participant concerns.

## Data collection

The household survey comprises five sections: (1) demographics and socioeconomic profile; (2) current and past health conditions; (3) well-being and lifestyle; (4) anthropometric measurements and; (5) screening for AF. The demographic and socioeconomic profile will be collected using the Sri Lanka Demographic and Health Survey 2016 questionnaire which includes education levels and occupation.[24] Current medical conditions and medical history of comorbid diseases will be collected using the Charlson Comorbidity Index[25] along with the recording of any current medications prescribed by physicians or indigenous healers. To assist participant's recalling their medical history, the data collection team will advise them to review their clinical books which is a paper notebook provided by the health system and commonly kept by patients who see clinicians for long-standing conditions in which clinical appointments, clinical assessments, investigations and medications are often detailed. Well-being and lifestyle assessment will be collected using the following validated instruments: General Anxiety Disorder-2 (GAD-2) scale,[26] Patient Health Questionnaire-2 (PHQ-2),[27] International Physical Activity Questionnaire—Short Form,[28] Audit C for alcohol use,[29] WHO STEPS instrument[30] for assessing

tobacco use, Berlin Questionnaire for sleep quality,[31] EQ-5D-5L (EuroQol-5 dimensions-5 levels) for quality of life,[32] and the Fried Frailty Index.[33] Diet will also be assessed by asking questions on consumption levels of relevant food groups.[34] Participants will be asked to describe their usage of healthcare services in the preceding year, including with public and private services and indigenous healers. Anthropometric measurements will include the recording of each participant's weight, height, waist and hip circumference along with their blood pressure (measured by data collectors). The household survey will take approximately 45 min per participant.

Participants will have their heart rhythm recorded using fingertip electrodes which activate a bipolar single-lead (lead 1) ECG recording through a smartphone application (AliveCor) that uses an automated algorithm to interpret the cardiac rhythm and identify AF.[35] AliveCor is a validated tool used internationally and is Food and Drug Administration approved.[36 37] Most studies have found the sensitivity and specificity of AliveCor to be 90% or above.[36] The application logs the participants anonymously, assigning a unique ID to each participant and stores a report of the rhythm detected. The participant's unique ID is entered into the data collection tool to enable linkage of the participant's heart rhythm results and their questionnaire data. Reports of the single-lead recording (AliveCor report) will be downloaded and reviewed by a consultant cardiologist off-site who will determine the outcome of the test. Any participant with a positive or undifferentiated AliveCor results will be referred to see the study consultant cardiologist and have a 12-lead ECG and relevant assessment. A member of the research team will follow-up with participants that had not visited the hospital within 3 months following referral. Patients access services through the existing government health service.

Digital data collection tools that had been developed and successfully deployed in Sri Lanka were adapted for use in this study.[38] We improved data quality and standardised terminology for diagnosis and comorbidities and included automatic data-logic checks for numerical values to reduce data entry errors. The final tool, using the REDCap (Research Electronic Data Capture)[39] software, includes satellite mapping and inbuilt date and time stamps and supports study coordinators with monitoring data collection and feedback. Data collected is automatically transferred to analytical dashboards (figure 1); thus, the study research team has in near-real time access to study recruitment, data quality, reasons for failure to consent and feasibility of study components. The digital tool is accessible via a mobile device and includes offline functionality, enabling the ability to overcome any issues with mobile data coverage.

The questionnaire and study tools were designed in English and then translated to Tamil by two independent translators. These translations were then reviewed by a working group of bilingual stakeholders from a range of academic, education and professional backgrounds to

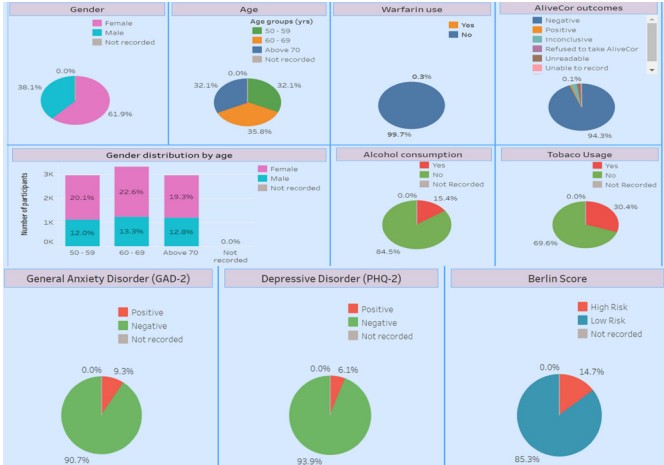

**Figure 1** Analytical online dashboard. A breakdown of demographics, lifestyle and well-being measures for participants recruited and consented thus far. PHQ-2, Patient Health Questionnaire-2.

ensure that the translation and phrasing of the questionnaire was appropriate for monolingual participants. The refined Tamil version of the questionnaire was then back translated to English and reviewed for integrity and retention of information by the study team.

For every cluster GN, a community volunteer will facilitate the data collection by introducing the data collectors to participants. Each data collection team will consist of a trained data collector and the community volunteer. Data collectors will be trained by researchers and community physicians from the University of Jaffna in study participation, consent process, administering the questionnaire and data entry using the digital data collection tool. Data collectors will also be trained in how to administer AliveCor for detecting AF. Trained data collectors will provide information to the participant regarding the study, and obtain written informed consent.

## Statistical methods

A direct standardisation technique will be used to obtain province-level prevalence data from the district level results. The proportion of the population exposed to known risk factors of AF will be calculated. Prevalence of risk factors in participants with established, without and with newly diagnosed AF will be determined. Multivariate logistic regressions will be used to calculate adjusted ORs for AF risk factors. Population attributable fractions will be calculated for risk factors that were included in the logistic regression model. Statistical analysis would be carried out using SPSS V.23 (IBM Corp. 2015; Armonk, New York, USA). Raw data will be reported along with scores from assessment tools (eg, GAD-2 and PHQ-2), derived using published definitions. All calculations will be published with open access, ensuring reproducibility of study findings.

## Study monitoring

Data quality (completeness, accuracy, logic) will be reviewed daily by the study's data quality coordinators from a central monitoring platform. The study will adhere to the FAIR (Findability, Accessibility, Interoperability, and Reusability) guiding principles of data management and sharing.[40] In-built dashboards will provide accrued metadata on screening for AF, eligibility, participant recruitment, data quality and prevalence of AF and other non-communicable diseases in the study population.[38 41] Weekly debriefs between the study team and data collectors will be used to discuss progress and address any challenges arising. In addition, source data will be verified in 1% of the questionnaires at random each month where research fellows will join the data collectors and observe study conduct and identify any potential biases in data collection which may impact inter-rater reliability.

## Data storage and security

Data will be stored on institutional network drives of the Information and Communication Technology Agency Sri Lanka, adhering to national data storage requirements.[42] Access to records and data will be limited to study personnel. All participant data will be anonymised by removing any identifiable data from the data collected in field.[43] The data may be linked using a serial number.

## Patient and public involvement

Community engagement workshops with key stakeholders were conducted by the study team. Medical Officers of Health and Public Health Midwives within each GN division were asked to participate and aid in the community engagement activities. Stakeholders included patient representatives and primary healthcare physicians, community health workers and specialist physicians working in the Northern Province. Stakeholders were asked to give their opinions on the research objectives and their perceptions of potential challenges of conducting large scale epidemiological studies in the province. This included how to achieve participant engagement from marginalised communities, improve health literacy and health-seeking behaviours and how to optimise data quality and impact population health policy.[44] These insights were used to inform the selection of study tools for exploring health-seeking behaviours, lifestyle assessment and participant information. Similar workshops are planned at 3-month intervals during participant enrolment, data collection and analysis, whereby emerging enablers and barriers to the study will be disseminated to stakeholders. Thus, we will create a dialogue whereby perceptions of current services can be evaluated and priorities for improvement be identified and agreed with stakeholders throughout the study. Participants from the stakeholder group will be asked to join a smaller working group who will guide the development of health education materials pertaining to lifestyle, general well-being and AF-specific management. They will provide feedback on the materials, their acceptability, potential barriers to

the adoption of new knowledge by the Northern Province community and codesign future implementation methods of the materials.

## COVID-19 pandemic

While the pandemic was emerging in Europe and the USA, COVID-19 cases were not yet present in Sri Lanka. We used this opportunity to discuss perceptions of the disease at the community engagement workshops, assess risk and adapt data collection procedures to reduce risk of transmission. Preventive measures such as hand washing techniques, maintaining social distancing and wearing face masks were explained to data collectors. Hand sanitiser was distributed to all data collectors and they were instructed to disinfect their hands frequently.

## ETHICS AND DISSEMINATION

The study received approval from the Ethics Review Committee of Faculty of Medicine, University of Jaffna (J/ERC/19/103/NDR/0208). Written informed consent will be obtained from all participants. A helpline for participants and family members and caretakers of the participants will be set up for any questions or concerns throughout and beyond the study period.

Any participant that is newly diagnosed with AF through this study will be referred to see the study consultant cardiologist. All participants and their families will be invited to use a purpose developed helpline, specifically established for this cohort. The helpline provides advice for participants and their families regarding health, well-being, ongoing comorbidity in addition to management of AF. Additionally, research fellows are trained to provide information and advice on lifestyle behaviours where appropriate, and where the participant should go for medical help for any health-related issues.

Results of the study will be made available to participants and/or their caregivers, as appropriate. The findings will be disseminated through abstract presentations at national and international conferences and a manuscript publication in a peer-reviewed journal.

## DISCUSSION

This study will be the first of its kind in Sri Lanka and one of the few community-based AF screening studies conducted in South Asia.[2] Our findings will determine the prevalence of AF within the Sri Lankan community and enable early and effective treatment for those newly diagnosed with AF. Hospital data in Sri Lanka and other LMICs are often limited, is of poor-quality and typically does not represent the true burden of conditions due to various reasons (healthcare-seeking behaviours, lack of diagnostic equipment in primary care, inefficient referral systems, patients lost to follow-up, etc).[45–47] This study will also highlight key enablers and barriers of implementing a population-based screening programme for AF in an LMIC setting, facilitating other LMICs with the means to adapt and implement similar programmes. Dissemination of these findings will provide evidence for policymakers and healthcare professionals of the true burden of AF within the communities they serve, enabling a discourse on how to develop effective care and management to reduce incident AF and improve timely diagnosis of existing AF.

Further to the study aims, recruitment of the 10 000 participants will form a distinct cohort of community members that can be used in future studies. All participants have been asked for consent to conduct further analysis with the data collected and to be contacted for future trials and surveys regarding AF and other non-communicable diseases. To our knowledge, no AF community-based cohort currently exists in South Asia. This study along with future studies carried out within this cohort will enhance the epidemiological evidence on non-communicable diseases in Sri Lanka. Subsequently, interventions can be developed and tested for improving the health and well-being of LMIC occupants, including those most vulnerable and disadvantaged.

The strengths of this study include the use of mHealth technology for screening for and diagnosing AF in an LMIC setting. In high-income countries the tool has been shown to be easy to use, cost-effective and yields a high accuracy rate.[36] Success of using this tool within the study will indicate the feasibility of using it in other resource-limited settings (including post-war settings) either in the community or in primary care settings that lack appropriate equipment. Conducting this study in an LMIC setting will provide important learnings for future screening programmes in LMICs, extending the current evidence beyond high-income countries. The use of analytical dashboards is another strength of the study. In near-real time, data is able to be reviewed and assessed by the study team both nationally and internationally for any quality issues, enabling immediate rectification and ongoing monitoring. The dashboards also allow for a unique opportunity to review the study findings as data collection progresses. There are however some limitations to be mentioned. One major limitation will be AliveCor's inability to detect paroxysmal AF; however, if feasible and acceptable, the device will offer an opportunity to increase rapid detection of AF in community settings of Sri Lanka. Due to the cross-sectional design of this study, only prevalence of AF will be feasible; therefore, AF incidence in the community will remain unknown. Another limitation due to the study design is the inability to infer on the causal relationship between risk factors and AF. Though, the collection of a wide range of demographic and lifestyle data will allow for determination of which risk factors are associated with an existing and new AF diagnosis.

In conclusion, our study will identify the prevalence of AF in the Sri Lankan general population. Further to this, our findings will enable learnings for screening programmes for other LMICs and provide evidence on the feasibility of using an mHealth tool for diagnosing

AF in a resource-limited setting. Such learnings have the potential to improve early diagnosis and prevention of AF in LMICs, which will increasingly become a global health priority.

**Author affiliations**
[1]Department of Community and Family Medicine, University of Jaffna, Jaffna, Northern Province, Sri Lanka
[2]National Intensive Care Surveillance-Mahidol Oxford Tropical Medicine Research Unit, Jaffna, Sri Lanka
[3]Department of Public Health and Epidemiology, University of Birmingham, Birmingham, UK
[4]Liverpool Centre for Cardiovascular Science, University of Liverpool, Liverpool, UK
[5]Department of Cardiology, Jaffna Teaching Hospital, Jaffna, Sri Lanka
[6]Centre for Inflammation Research, The University of Edinburgh, Edinburgh, UK

**Acknowledgements** The authors would like to thank the Medical Officers of Health, Public Health Midwives, data collectors and volunteers for their dedication and invaluable assistance with the pilot study and with recruitment thus far.

**Collaborators** Ajini Arasalingam, Isabela M Bensenor, Peter Brocklehurst, Kar Keung Cheng, Wahbi El-Bouri, Mei Feng, Alessandra C Goulart, Sheila Greenfield, Yutao Guo, Gustavo Gusso, Lindsey Humphreys, Kate Jolly, Sue Jowett, Emma Lancashire, Deirdre A Lane, Xuewen Li, Yan-guang Li, Trudie Lobban, Paulo A Lotufo, Semira Manaseki-Holland, David Moore, Rodrigo D Olmos, Elisabete Paschoal, Paskaran Pirasanth, Carla Romagnolli, Itamar S Santos, Alena Shantsila, Vethanayagan Antony Sheron, Isabelle Szmigin, Meihui Tai, Timo Tolppa, Ana C Varella, Hao Wang, Jingya Wang, Hui Zhang, Jiaoyue Zhong.

**Contributors** BK, GNT, KN, GL, MG, RS, SK, RH, AB and KS conceptualised and designed the study. VAS, PU, CK, SK, BK, RS and KS contributed to the piloting and implementation of the study. SK, TEG and AB drafted the manuscript. All authors reviewed and approved the final manuscript for submission.

**Funding** This work was supported by the National Institute for Health Research (project reference 17/63/121, NIHR Global Health Research Group on Atrial Fibrillation Management).

**Competing interests** GL is a consultant and speaker for BMS/Pfizer, Boehringer Ingelheim and Daiichi-Sankyo; no fees are received personally.

**Patient and public involvement** Patients and/or the public were involved in the design, or conduct, or reporting, or dissemination plans of this research. Refer to the Methods section for further details.

**Patient consent for publication** Not applicable.

**Provenance and peer review** Not commissioned; externally peer reviewed.

**ORCID iDs**
Shribavan Kanesamoorthy http://orcid.org/0000-0002-9242-9503
Graham Neil Thomas http://orcid.org/0000-0002-2777-1847
Krishnarajah Nirantharakumar http://orcid.org/0000-0002-6816-1279
Gregory Y H Lip http://orcid.org/0000-0002-7566-1626

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
