## [Reviewer comments · BMJ Open]

ARTICLE DETAILS

TITLE (PROVISIONAL)	Prevalence of atrial fibrillation in Northern Sri Lanka: A study protocol for a cross-sectional household survey
AUTHORS	Kanesamoorthy, Shribavan; Sheron, Vethanayagam Antony; Uruthirakumar, Powsiga; Kodippily, Chamira; Kumarendran, Balachandran; Gooden, Tiffany; Thomas, G. Neil; Nirantharakumar, Krishnarajah; Lip, Gregory; Guruparan, Mahesan; Haniffa, Rashaan; Surenthirakumaran, Rajendra; Beane, Abi; Subaschandran, Kumaran

VERSION 1 – REVIEW

REVIEWER	Charlotte Hespe The University of Notre Dame Australia, School of Medicine
REVIEW RETURNED	14-Jan-2022

GENERAL COMMENTS	Thank you for submitting this protocol paper for review. The study is extremely interesting and you have adopted a really novel approach to gathering the data. The paper is well structured and easy to read. However, I have several queries that I would like you to address in the paper with respect to the study design and addressing some key risks posed by the study that are currently not discussed in your paper. 1. Can you please add in an explanation for your choice of 50 years old and over for the selection of the participant in the study. What evidence or data was this decision based upon? I note your mention on Page 8 lines 22- 34 regarding the screening age of 65 internationally but that LMIC countries have increasing numbers of CVD deaths in the "younger population". This would be a good place to add in reasons for the selection of age 502. Participant data. The data collection for the study is very comprehensive with a large number of validated surveys and questionnaires for each participant to complete. You include a section about the use of a Pilot study to inform the finalisation of the design for your study. This section includes that the participants were asked about the acceptability of the questionnaires. However, there is no mention of what the feedback stated nor how the study was adapted in response to the feedback. I have personal concerns, that the data collection is a very lengthy process and so would appreciate you including comments about the feedback by participants regarding the process of data collection. Overall estimation of time taken to complete all of the data collection would be very helpful to the readers to increase our understanding of the complexity of the role for the data collectors as well as the participants.3. There is no mention of the use or reporting of the information gained from these surveys. This raises some significant ethical
--

	health and safety concerns. A. There is a figure included that shows the results from all of the current participants. This demonstrates there are quite a few participants who have "high risk" or positive scores recorded. Yet there is no information provided in your protocol about responding to these results. There is no mention of any care or irresponsibility being taken by the data collection team / nor the review team after the identification of a mental health / dietary or alcohol-related health issue? Is this an oversight? It is essential that you include information about what and how you are responding to these results - over and above the AF prevalence findings. B. You state that when a participant is identified as having newly diagnosed AF they will be referred to the local Cardiologist for review. Could you please also include information about the undifferentiated results? Can you also please include details about clinical governance for the follow up of these results - what is the time frame / who covers the costs? / Is there any expected costs for the participants with respect to further investigations/medications or travel to see the specialists? C. Are the research team planning on using the demographic data for further papers or research? It would be an opportunity missed if not. Can you please include a sentence stating what the intention for use of this data is to be and also if this is included in the consent process for the participants? 4. You mention in the discussion section that you are planning to use this cohort for further investigations and research. Has this been included in the consenting process? 5. The authors state that this method of data collection is potentially providing information for LMIC screening programs in the future. Could you please clarify this statement as I do not believe that this is either an objective or a measurable finding that will be achieved from this study.
--	---

REVIEWER	Josep Lluís Clua-Espuny
REVIEW RETURNED	16-Jun-2022

GENERAL COMMENTS	Se trata de un estudio innovador y de alta potencialidad, pero insuficiente metodológicamente. Especialmente a destacar los siguientes puntos  1. Las fechas del estudio 06/20 a 03/22: si bien es un estudio no terminado, estaria muy cerca de la finalización de la recogida de datos, por lo que el editor debería evaluar este hecho como condición. 2. Si el objetivo principal es "la prevalencia de la fibrilación auricular", por que incluir en los objetivos "other noncommunicable diseases". Si se incluye, deberían especificarse cuales. 3. La metodología en la definición de la muestra y la selección aleatoria debería describirse más detalladamente: poder de la muestra errores alfa y beta, etc... y los motivos de exclusion. 4. Por que a partir de 50 años y o de 65 como aconsejan las Guías aportadas como referencia? 5. Finalmente, el punto debil que considero de mas importancia es la validación de los resultados del Alive-Cor: no se explica con el suficiente detalle como se interpretan los resultados, cuál es el protocolo en caso de no ser evaluable el registro, cómo se validan definitivamente los posibles resultados positivos (se realiza ECG?, se hace seguimiento, se instaura TAO, se refiere a medico referencia, etc...?), cómo se abordan los posibles casos de FA paroxística? que protocolo se sigue con casos ya diagnosticados de
--

	FA? My best wishes,
--	------------------------

VERSION 1 – AUTHOR RESPONSE

Reviewer 1's comments

Comment 1: Thank you for submitting this protocol paper for review.

The study is extremely interesting and you have adopted a really novel approach to gathering the data. The paper is well structured and easy to read.

However, I have several queries that I would like you to address in the paper with respect to the study design and addressing some key risks posed by the study that are currently not discussed in your paper.

Response 1: Many thanks for the positive assessment and kind words. We appreciate the time you have taken to review and comment on our manuscript. We have responded to your suggestions and comments and updated the manuscript accordingly, as mentioned below.

Comment 2: Can you please add in an explanation for your choice of 50 years old and over for the selection of the participant in the study. What evidence or data was this decision based upon? I note your mention on Page 8 lines 22- 34 regarding the screening age of 65 internationally but that LMIC countries have increasing numbers of CVD deaths in the "younger population". This would be a good place to add in reasons for the selection of age 50

Response 2: In many middle- and high-income countries, the screening age of atrial fibrillation is 65 but similar studies to ours have chosen age 50 as the screening age limit due to the increasing numbers of CVD deaths in the younger populations in LMICs (examples include Soni et al 2018, Davidson et al 2022 and Stavrakis et al 2021). We have amended the manuscript to make this clearer (as below).

Introduction: "Such practice remains limited in low- and middle-income countries (LMICs), where cardiovascular diseases in the younger populations are rapidly becoming the leading cause of mortality.[11] As a result, literature increasingly supports an age threshold of 50 years and above for AF screening.[12,13,14]"

Methods: "Consistent with other LMIC studies,[12] individuals aged 50 years or above will be considered eligible; additionally, they must be fluent in the Tamil language."

Comment 3: Participant data. The data collection for the study is very comprehensive with a large number of validated surveys and questionnaires for each participant to complete. You include a section about the use of a Pilot study to inform the finalisation of the design for your study. This section includes that the participants were asked about the acceptability of the questionnaires. However, there is no mention of what the feedback stated nor how the study was adapted in response to the feedback.

Response 3: A few key responses from pilot participants resulted in minor changes to the data collection procedure, but mainly made us more aware of how to improve data quality. For instance, when we were collecting data about the pilot participants' comorbidities, participants said that they had diseases that could not be confirmed due to the absence of their clinic books. As a result, we decided to record this information and explicitly mark which diseases were confirmed by their clinic book and which were not. Additionally, at the start of the pilot, we asked participants to complete the questionnaire about the medications they were taking but participants found this time consuming and difficult to do. To improve this, we grouped the medications together in categories and had the data collector mark them instead of the participant. As already mentioned in the manuscript, "Recurring themes regarding concerns over the reasons for conducting the survey and potentially being disadvantaged if they were found to have AF prompted a review of participant information leaflets and further training for the data collection team". In addition to this, some pilot participants expressed concerns that the device may take their fingerprint which could then be used for other purposes. As a response, we trained our data collectors in how to respond to such concerns and how to explain to the participant that this was not the case. This was all the feedback we received that resulted in adaptations of the study. We have added this information into our manuscript, as below.

"Feedback following the pilot revealed i, data collector concerns regarding availability of and access to clinic books for relevant medical history, ii, participant concerns regarding future use of fingerprints, reasons for the survey and being disadvantaged if they were found to have AF, and iii, participants understanding of medical prescriptions. In response, we made the following adaptations to the study conduct protocol prior to commencing data collection: reporting of medical history will be reported as either being directly observed from clinic books or via participant recall; medications will be reported by categories for simplicity and; an update of participant information leaflets to insure the aim of the study, data security and what happens after the study is clear. Additionally, further training for the data collection team was provided on the management of personal information, the AliveCor device and how to respond to participant concerns."

Comment 4: I have personal concerns, that the data collection is a very lengthy process and so would appreciate you including comments about the feedback by participants regarding the process of data collection. Overall estimation of time taken to complete all of the data collection would be very helpful to the readers to increase our understanding of the complexity of the role for the data collectors as well as the participants.

Response 4: A household visit will take 45 minutes per participant. We have included this information into our manuscript, as below.

“The household survey will take approximately 45 minutes per participant.”

Comment 5: There is no mention of the use or reporting of the information gained from these surveys. This raises some significant ethical health and safety concerns.

A. There is a figure included that shows the results from all of the current participants. This demonstrates there are quite a few participants who have "high risk" or positive scores recorded. Yet there is no information provided in your protocol about responding to these results. There is no mention of any care or irresponsibility being taken by the data collection team / nor the review team after the identification of a mental health / dietary or alcohol-related health issue? Is this an oversight? It is essential that you include information about what and how you are responding to these results - over and above the AF prevalence findings.

Response 5: Apologies for this oversight, as we do have a system in place for responding to the medical information identified from the questionnaires; these are described as follows and are now included in our manuscript (as below). We have established a helpline service specifically for the participants and their families to call if they should have any questions (AF related or not). Through the helpline, we deliver telemedicine services and a consultant family physician talks directly to participants that call and they guide them for any necessary further action. Furthermore (but not mentioned in the manuscript), following the data collection described in our protocol, we plan to follow up with all consenting participants for collecting blood samples; at this time, they will meet directly with a doctor which will provide them with a face-to-face interaction and enable the doctor to guide them further regarding any health-related issues. The research fellows (i.e. data collectors) were trained in providing information on lifestyle behaviours and on where the participant should go for medical help for any unattended health-related issue.

“All participants and their families are invited to use a purpose developed helpline, specifically established for this cohort. The helpline provides advice for participants and their families regarding health, wellbeing, ongoing comorbidity in addition to management of AF. Additionally, research fellows are trained to provide information and advice on lifestyle behaviours where appropriate, and where the participant should go for medical help for any health-related issues.”

Comment 6: B. You state that when a participant is identified as having newly diagnosed AF they will be referred to the local Cardiologist for review. Could you please also include information about the undifferentiated results? Can you also please include details about clinical governance for the follow up of these results - what is the time frame / who covers the costs? / Is there any expected costs for the participants with respect to further investigations/medications or travel to see the specialists?

Response 6: All results of the AliveCor will be sent to the local cardiologist for review. The data collection team (supported by medical research fellows) will be instructed to refer any participants to the nearest hospital to have a 12-lead ECG conducted for confirmation or clarification of the AliveCor results. A research fellow will regularly monitor the follow up care/visits and members of the helpline team will call any participants who had not visited the hospital within 3 months of referral to remind them to visit the hospital and answer any questions they may have. Healthcare access for cardiologists is free at point of care. We have added the below texts into our manuscript:

“Reports of the single lead recording will be downloaded and reviewed by a consultant cardiologist off-site. Any participant with a positive or undifferentiated AliveCore results will be referred to see the study consultant cardiologist and have a 12-lead ECG and relevant assessment. A member of the research team will follow-up with participants that had not visited the hospital within 3 months following referral. Patients access services through the existing government health service.”

“Publicly available healthcare including medical appointments, medications and medical procedures are free to all Sri Lankan citizens.”

Comment 7: C. Are the research team planning on using the demographic data for further papers or research? It would be an opportunity missed if not. Can you please include a sentence stating what the intention for use of this data is to be and also if this is included in the consent process for the participants?

Response 7: Yes, we have mentioned this in the discussion, as below.

“Further to the study aims, recruitment of the 10,000 participants will form a distinct cohort of community members that can be utilised in future studies. All participants have been asked for consent to conducting further analysis with the data collected and to being contacted for future trials and surveys regarding AF and other noncommunicable diseases. To our knowledge, no AF community-based cohort currently exists in Southeast Asia. This study along with future studies carried out within this cohort will enhance the epidemiological evidence on noncommunicable diseases in Sri Lanka. Subsequently, interventions can be developed and tested for improving the health and wellbeing of LMIC occupants, including those most vulnerable and disadvantaged.”

Comment 8: You mention in the discussion section that you are planning to use this cohort for further investigations and research. Has this been included in the consenting process?

Response 8: Yes, we have amended the text to make this clearer, please see response 7 above.

Comment 9: The authors state that this method of data collection is potentially providing information for LMIC screening programs in the future. Could you please clarify this statement as I do not believe that this is either an objective or a measurable finding that will be achieved from this study.

Response 9: We do not intend to provide information for LMIC screening as an objective or measurable aim of this study. However, we mentioned in the manuscript that this study will provide first-time evidence on “key enablers and barriers of implementing a population-based screening programme for AF in an LMIC setting”, thus, we will be providing other LMICs with the tools to adapt and implement similar programmes. Whilst this is not a specific aim of our study, we feel it is nonetheless important to note as a strength of the study, as mentioned in the discussion: “Success of using this tool within the study will indicate the feasibility of using it in other resource-limited settings (including post-war settings) either in the community or in primary care settings that lack appropriate equipment. Conducting this study in an LMIC setting will provide important learnings for future screening programmes in LMICs, extending the current evidence beyond high-income countries.” We feel our objective and measurable aim of the study (i.e. to determine the prevalence of AF and other noncommunicable diseases in Northern Province, Sri Lanka and to evaluate the feasibility and acceptability of smartphone enabled single lead electrocardiogram (ECG) for the purpose of diagnosing AF) is clear throughout the manuscript and we hope you agree with this assessment.

Reviewer 2's comments

Comments to the Author: PLEASE NOTE THAT THE REVIEW WAS SUBMITTED IN SPANISH - THIS TRANSLATION BY GOOGLE TRANSLATE (ORIGINAL TEXT FOLLOWS)

Comment 1: This is an innovative study with high potential, but methodologically insufficient. Especially note the following points

Response 1: Many thanks for taking the time to review our paper and provide important and helpful suggestions. We hope our responses below and amendments to our manuscript address your concerns regarding the methods.

Comment 2: The dates of the study 06/20 to 03/22: although it is an unfinished study, it would be very close to the end of data collection, so the editor should evaluate this fact as a condition.

Response 2: It is important to note that we submitted our protocol manuscript to BMJ Open in August 2021, well before the expected end date of the study (March 2022). Unfortunately, the journal experienced delays in getting our manuscript reviewed, hence why we did not receive your comments

until after the expected study end date. However, the study is still not finished and we have experienced delays to data collection due to COVID and the current economic crisis and civil unrest in Sri Lanka. We now suspect that the study end date will be October 2022. We have updated this date within the manuscript.

(We note this reviewer comment is for the editor. Please note, this article is submitted as a protocol/methods paper. No conclusion is being sought.)

Comment 3: If the main objective is "the prevalence of atrial fibrillation", why include "other noncommunicable diseases" in the objectives? If included, which should be specified.

Response 2: Our intentions are to identify the prevalence of AF and associated risk factors, many of which are other non-communicable diseases. We have amended the text within the manuscript to reflect this, as below.

"This study aims to implement an AF screening programme using smartphone technology, feasible for delivery at scale in resource constrained LMIC settings. Specifically, we aim to determine the prevalence of AF and associated risk factors in Northern Province, Sri Lanka and to evaluate the feasibility and acceptability of smartphone enabled single lead electrocardiogram (ECG) for the purpose of diagnosing AF.

Comment 4: The methodology in the definition of the sample and the random selection should be described in more detail: power of the sample, alpha and beta errors, etc...

Response 4: The sample size was calculated using the formula $n = z^2 \times P(100-P)/d^2$. We assumed a priori the prevalence of AF (P) to be 1%; this assumption is based on Sri Lankan study that reported the prevalence of stroke and associated risk factors including AF in the Western Province. To adjust the required sample size for cluster sampling, we considered a design effect of 2. We increased the sample size by 10% to account for non-participation. Based on the above calculations and assumptions, the final sample size was calculated as 10,000. The alpha level was 0.05 and beta level was 0.2 which provided our required sample size of 10,000 a power of 0.8. We have added this information into our manuscript, the text now reads as follows:

"For the sample size calculation, the prevalence of AF in Sri Lanka was estimated to be 1% based on previous evidence.[2, 17] We used a design effect of 2 to adjust for cluster sampling, and we used an alpha level of 5% and beta level of 20%. We increased our sample size by 10% to account for non-participation,[18] leading to a minimum sample size of 10,000 participants required."

Comment 5: ...and the reasons for exclusion.

Response 5: Based on already published methods (Patino et al 2018), participants suffering from terminal illnesses, life-threatening medical conditions or any one in need of immediate hospital admission were excluded from this study.

“Based on already published methods,[15] anyone suffering from terminal illnesses, in need of immediate hospital admission or currently an inpatient in a hospital setting will be excluded.”

Comment 6: Why from 50 years old and or from 65 as recommended by the Guides provided as a reference?

Response 6: This has been addressed (please see comment 2 by reviewer 1 and our corresponding response).

Comment 7: Finally, the weak point that I consider most important is the validation of the AliveCor results: it is not explained in sufficient detail how the results are interpreted, what the protocol is if the registry is not evaluable, how Are the possible positive results definitively validated (is an ECG performed?),

Response 7: All participants with an AliveCor inconclusive or positive result will be referred to the hospital to take a 12-lead ECG to confirm or clarify the AliveCor results. A consultant cardiologist will review all AliveCor results and make the appropriate recommendations to refer the patient or not. As mentioned in the methods section of the paper, “AliveCor is a validated tool used internationally and is FDA approved.[32, 33] Most studies have found the sensitivity and specificity of AliveCor to be 90% or above.[32]”; therefore, AliveCor has been validated in previous studies and is expected to have high sensitivity and specificity in our study.

Comment 8: is follow-up done, is TAO established, is the referral physician referred, etc...?),

Response 8: In our described study, follow-up was only done for participants with a positive or unclear AliveCor result, in that they were referred to take a 12-lead ECG at the hospital. This decision was made by the cardiologist consultant that reviewed the AliveCor readings for each participant. If

AF was confirmed by the 12-lead ECG, they were referred to the relevant healthcare clinic for specialised AF care (i.e. to monitor anticoagulation medication and INR), as per standard practice in Sri Lanka. We have added text to the manuscript to clarify this (see text below). Not included in this protocol are the plans, methods and ethical approval for a separate study to follow up all participants (given consent) to collect blood samples and further data related to AF and other noncommunicable diseases. Apologies, we are unsure what you mean by TAO.

“Reports of the single lead recording will be downloaded and reviewed by a consultant cardiologist off-site. Any participant with a positive or undifferentiated AliveCore results will be referred to see the study consultant cardiologist and have a 12-lead ECG and relevant assessment. A member of the research team will follow up with participants that had not visited the hospital within 3 months following referral.”

Comment 9: How are possible cases of paroxysmal AF addressed?

Response 9: This is an important point as this is a limitation of the study. The AliveCor is unable to identify paroxysmal AF and therefore anyone with paroxysmal AF will be missed from our study. We have added this as a limitation in the discussion of our paper, as below.

“One major limitation will be the AliveCor’s inability to detect paroxysmal AF; however, if feasible and acceptable, the device will offer opportunity to increase rapid detection of AF in community settings in Sri Lanka.”

Comment 10: What protocol is followed with already diagnosed cases of AF?

Response 10: Participants that had a confirmed diagnosis of AF prior to data collection will be advised to continue to follow their routine care provided by their current doctor.

VERSION 2 – REVIEW

REVIEWER	Charlotte Hespe The University of Notre Dame Australia, School of Medicine
REVIEW RETURNED	12-Sep-2022

GENERAL COMMENTS	thank you for comprehensively addressing all the concerns noted in the first review of your article. I only have one remaining request - which is to provide a slightly longer statement regarding the choice of screening for the 50s and over rather than 65s in your paper. I feel it would be more helpful to readers to know that current recommendations are for 65 and over but due to emerging evidence
---

	about the higher risk in your community for a younger cohort, you have made this choice. It reads as if the current recommendations are for the 50s and over rather than emerging evidence showing there might be a gap if only doing for 65s and over. Your data may then be useful in assisting in changing International guidelines.
--	---

REVIEWER	Josep Lluís Clua-Espuny
REVIEW RETURNED	23-Aug-2022

GENERAL COMMENTS	Just three short comments: 1/ It is going to be a hard job for a sample of 10000 individuals 2/ You wrote "any participant with a positive or undifferentiated AliveCore results will be referred to see the study consultant cardiologist....." but who is going to diagnose the atrial fibrillation and what criteria are going to be used to? 3/ I consider the possible exclusion criteria not enough because you will need access to medical history and you wrote in Discussion about the poor-quality of hospital data " and typically does not represent the true burden of conditions"
---

VERSION 2 – AUTHOR RESPONSE

Reviewer 2

Comment 1

It is going to be a hard job for a sample of 10000 individuals

Response 1

We have already allocated the resources, trained data collectors and funds required to complete this task. Through the pilot, we confirmed the feasibility of data collection which is now underway and showing promising results. Our online dashboard will be helpful in identifying any issues with reaching this target in near-real time.

Comment 2

You wrote "any participant with a positive or undifferentiated AliveCore results will be referred to see the study consultant cardiologist....." but who is going to diagnose the atrial fibrillation and what criteria are going to be used?

Response 2

A cardiologist located at the Jaffna Teaching Hospital will review all results from the AliveCor report to determine the outcome. We have updated our manuscript to make this more explicit, as below.

"Reports of the single lead recording (AliveCor report) will be downloaded and reviewed by a consultant cardiologist off-site who will determine the outcome of the test. Any participant with a positive or undifferentiated AliveCor result will be referred to see the study consultant cardiologist and have a 12-lead ECG and relevant assessment."

Comment 3

I consider the possible exclusion criteria not enough because you will need access to medical history and you wrote in Discussion about the poor-quality of hospital data " and typically does not represent the true burden of conditions"

Response 3

All data collectors (who include a medically trained doctor and two nursing graduates) will be trained and able to identify whether the person needs immediate hospital admission or has a terminal illness. We have added this information into the manuscript, as below.

“All data collectors – which includes a medically trained doctor and two nursing graduates – will be trained to assess individuals for eligibility.”

Reviewer 1

Comment 1

thank you for comprehensively addressing all the concerns noted in the first review of your article. I only have one remaining request - which is to provide a slightly longer statement regarding the choice of screening for the 50s and over rather than 65s in your paper. I feel it would be more helpful to readers to know that current recommendations are for 65 and over but due to emerging evidence about the higher risk in your community for a younger cohort, you have made this choice. It reads as if the current recommendations are for the 50s and over rather than emerging evidence showing there might be a gap if only doing for 65s and over. Your data may then be useful in assisting in changing International guidelines.

Response 1

We agree this is an important point to highlight. We have added more information in the introduction and methods of the manuscript, as below.

Introduction: “Such practice remains limited in low- and middle-income countries (LMICs), where cardiovascular diseases are being increasingly reported in younger populations (ages 50 years and up), and which are rapidly attributing to causes of mortality.[11-14] Chronic disease prevalence in this relatively young population appears as a result of both childhood conditions such as rheumatic fever and valvular heart disease, and early onset of diabetes and hypertension as a result of age and lifestyle-related factors.[15]”

Methods: “Consistent with other LMIC studies and in response to the increasing literature on morbidity and chronic disease prevalence in South Asia, we selected a threshold of 50 years and above for inclusion in screening, a lower than current recommended practice of 65 years.[12] Therefore, individuals aged 50 years or above will be considered eligible; additionally, they must be fluent in the Tamil language.”